# Food safety knowledge, attitudes, and practices among Jordanian women handling food at home during COVID-19 pandemic

Tasneem M. Al-Jaberi[1], Anas A. Al-Nabulsi[1], Tareq M. Osaili [1,2]*, Amin N. Olaimat[3], Sawsan Mutlaq[1]

1 Department of Nutrition and Food Technology, Faculty of Agriculture, Jordan University of Science and Technology, Irbid, Jordan, 2 Department of Clinical Nutrition and Dietetics, College of Health Sciences, University of Sharjah, Sharjah, United Arab Emirates, 3 Department of Clinical Nutrition and Dietetics, Faculty of Applied Medical Sciences, The Hashemite University, Zarqa, Jordan

* tosaili@just.edu.jo

**Data Availability Statement:** All relevant data are within the paper.

**Funding:** The author(s) received no specific funding for this work.

## Abstract

Concerns over food safety issues during the coronavirus pandemic (COVID-19) have sparked worldwide interest. Being part of a farm-to-fork food safety chain, food handlers at home are the final line of defense in reducing foodborne diseases. The present study used a cross-sectional survey to investigate the knowledge, attitudes, and practices (KAP) of women food handlers in Jordan. The survey investigated the effect of the COVID-19 pandemic on women who handle food at home in terms of food safety KAP. One thousand one hundred twenty-six respondents completed a food safety questionnaire during the COVID-19 pandemic. With a mean score of 22.1 points out of 42, the results showed that women who handle food in their houses had insufficient knowledge, negative attitudes, and incorrect practices concerning food safety. The respondents demonstrated high knowledge, attitudes, and practices in the personal hygiene, cleaning and sanitation areas ($\geq$ 60.0%). On the other hand, participants' knowledge, attitudes, and practices regarding contamination prevention, health issues that would affect food safety, symptoms of foodborne illnesses, safe storage, thawing, cooking, keeping, and reheating of foods, as well as COVID-19 were all low (< 60.0%). The correlations between participants' total food safety KAP scores and education, age, experience, region, and the pandemic effect on food safety were statistically significant (P $\leq$ 0.05). To the best of our knowledge, this study is the first conducted in Jordan to investigate food safety knowledge, attitudes, and practices by women handling food at home during the COVID-19 pandemic.

## 1. Introduction

Concerns about food safety have been raised globally after the World Health Organization's (WHO) declaration of the COVID-19 pandemic in early 2020 as a worldwide crisis. This emergency inspired a global effort to adapt hygienic measures to prevent viral spread through respiratory droplets, generated via sneezing, coughing, or simply by breathing and talking [1].

**Competing interests:** The authors have declared that no competing interests exist.

During COVID-19, food safety was the most affected part of the food system as more precautions were added in each stage of food production [2]. Nonetheless, the pandemic restrictions have caused the majority of restaurants around the world to be locked down or remain functional only for take-away or delivery service. However, recent studies showed no indication that COVID-19 can be transmitted through food [2]. The studies found that Coronaviruses cannot proliferate on the surface of food packages because they require a living animal or human host to reproduce and thrive. Also, food packages do not need to be sanitized, but hands should be disinfected after touching them and before eating [3].

Food safety has always been a global dilemma for consumers, food providers, and government agencies due to foodborne pathogens. Pathogen-contaminated foods have always been a concern for global public health due to the diseases and disorders that may ensue from their ingestion. The WHO estimated that contaminated food causes almost 600 million disease cases and 420,000 deaths annually worldwide due to bacteria, viruses, parasites, poisons, and chemicals [4]. In addition, the Eastern Mediterranean has the third-highest rate of foodborne diseases per capita linked to preparing food with contaminated water, poor hygiene and inadequate conditions in food preparation and storage, lower levels of literacy and education and insufficient food safety knowledge or implementation of relevant regulations [4]. The incidence of foodborne infections in Jordan is unclear, although research has evaluated the burden of selected foodborne pathogens for one year. Jordan has recorded at least 6612 *Salmonella*, 16,260 *Shigella*, and 6912 *Brucella* outbreaks every year [5]. Recently in Jordan a dramatic food poisoning outbreak claimed the life of a 5-year-old child and affected over 800 individuals. They all had consumed low-cost shawarma from a restaurant on the outskirts of the capital, Amman [6].

According to the WHO, foodborne illnesses are caused by various factors, including cross-contamination, poor storage conditions, insecure food sources, food mishandling, poor personal hygiene, and improper cooking [7]. Foodborne outbreaks can occur anywhere, often during food handling, storage, and preparation [8]. Previous studies have found that food consumed at home is the leading source of foodborne illness outbreaks among consumers, but there is uncertainty since many instances are often misdiagnosed or underreported [9]. A cross-sectional study was done in Al-Ahsa, Saudi Arabia, among 239 Saudi women to assess their food safety knowledge and practices competence. The study included open-ended questions about several areas of food safety knowledge and practices at home and during shopping. The results of the food safety knowledge assessment revealed that almost half of the participants achieved a good score. In comparison, 75% of participants achieved a good score in food safety practices [10]. Another study was conducted among 1,000 women in Lahore households to measure their food safety knowledge, attitude, and practices in the home. The study found that most women at home have insufficient food safety knowledge with about 91% lacking the necessary food safety knowledge, and they fell short of adopting safe food handling practices [11].

Furthermore, according to an European Union overview study, 40.5% of foodborne outbreaks happen at home due to unhygienic manipulation of the ingredients by food handlers, cross-contamination, and temperature abuse [12]. To prevent foodborne illness at home, food handlers, especially women, need to have adequate knowledge and exhibit good practices in food preparation, handling, and storage to minimize cross-contamination and reduce hazards [13]. Many survey studies have indicated that food handlers in the home lack enough knowledge about food safety; however, none were specifically conducted on women who handle food at home [9, 13–16].

It is evident that there are few studies examining food safety on a national scale in Jordan. A study conducted in 2020 to assess the food safety knowledge of Jordanians indicated that food consumers in Jordan lack sufficient knowledge of food safety [17]. Our study is different from

the previous report in the populations examined, with only women participating in the current study and both women and men participating in the previous study. Also, there are differences in the range of food safety aspects covered in the two surveys. The current study covered almost all aspects of food safety, contrary to the previous work.

During the COVID-19 pandemic, food systems confronted significant obstacles in ensuring safe food handling and consumption along the food chain from farm to fork under the jurisdiction of relevant government agencies [18]. Therefore, there was an urgent need to examine knowledge, attitudes, and practices (KAP) regarding food safety during the era of the COVID-19 pandemic. In response several survey studies were conducted to investigate food safety KAP of food handlers among different sectors worldwide, although there were very few studies in Jordan. According to the findings, food safety KAP has improved during this public health crisis [19–23].

Most countries resorted to applying home quarantine to stop the spread of COVID-19, which increased homemade food preparation [24]. According to the societal culture in the Middle East area, women play a critical role in lowering the incidence of foodborne disease because they are primarily responsible for meal preparation at home [10]. As a result, this research aimed to evaluate the food safety KAP among women food handlers at home during the COVID-19 pandemic in Jordan. Work was also designed to investigate the impacts of the COVID-19 pandemic on food safety KAP.

## 2. Materials and methods

### 2.1 Research strategy

This cross-sectional study relied on an online, self-administered questionnaire and was conducted in 2021 to evaluate the KAP among Jordanian women handling food at home during the COVID-19 pandemic. The Jordan University of Science and Technology's Deanship of Graduate Studies and Institutional Review Board (IRB) authorized the study and procedure in session number 29/146/2021. The sample size of this study was calculated according to the latest data published by the Department of Statistics in Jordan estimating the number of households in Jordan to be 2,206,129 [25]. Therefore, a sample size of at least 1039 people would be necessary to have confidence in the results at the 99.99% level according to the "Sample Size Calculator," using the equations below.

$$CI = p \pm z \times \sqrt{\frac{p(1-p)}{n}}$$

Where $CI$ is Confidence Interval, z is z score, $p$ is the population proportion and $n$ is the sample size

$$Unlimited\ population: \ n = \frac{z^2 \times p(1-p)}{\varepsilon^2}$$

$$Finite\ population: \ n' = \frac{n}{1 + \frac{z^2 \times p(p-1)}{\varepsilon^2 N}}$$

Where $z$ is the z score, $\varepsilon$ is the margin of error, N is the population size and $p$ is the population proportion.

Hence, a total of 1126 women aged 18 years or older who had the primary responsibility of food preparation at home participated in the study. The participants were provided with the required information regarding the study and declared their agreement to participate through written consent at the beginning of the questionnaire.

## 2.2 Questionnaire design

The survey was split into three parts: the first part included queries on the socio-demographic data of women food handlers (age, marital status, educational level, and years of experience). The second and third parts included questions on KAP on food safety and COVID-19 aspects, which were adapted from previous papers with some improvement [18, 26, 27]. Personal hygiene, cleaning and sanitation, cross-contamination prevention, health issues that would affect food safety, safe storage, thawing, cooking, keeping, and reheating of foods, signs and symptoms of foodborne illnesses, and COVID-19 KAP were the topics covered in the second and third part.

## 2.3 Pilot study

The survey was translated from English to Arabic. After that, food safety experts individually reviewed it to authenticate survey content based on subject complexity and accurately assess what it was designed to measure. After that, a face-to-face validation of the questionnaire was done with 10 women who were chosen randomly. These participants were asked to keep note of the time it took them to fill out the questionnaire, how understandable the content was, and the language and vocabulary used. Food safety experts and academics examined and debated their replies to the questionnaire's instructions and content. In response to their feedback, minor revisions were made in the questionnaire. All questions were multiple-choice and based on Likert scale, which took between 15–20 min for each participant to complete.

Finally, to assess the questionnaire's reliability, a pilot study was carried out with 30 women, who were subsequently requested not to fill out the final version of the survey. The internal consistency reliability approach was used to determine the extent to which the food safety KAP test is reliable in assessing what it is supposed to evaluate. Regarding general internal reliability, the questionnaire was found to be reliable with a Cronbach's Alpha = 0.71.

## 2.4 Distribution of questionnaires and response bias

The questionnaire was published in groups made only for housewives via several platforms on social media to guarantee that each person from the target population had an equal likelihood of receiving a survey invitation. Since not all people have a social media account that they can use to respond to online surveys, the investigators handed out additional questionnaires after calling individuals who did not have a social media account. Researchers kept the survey anonymous and worked on creating a questionnaire with neutrally framed questions that did not offer leading response possibilities to eliminate bias.

## 2.5 Statistical analysis

The data were analyzed using the IBM SPSS Statistical Package for Social Sciences. For each variable, measures of dispersion (means, standard errors, frequencies, and percentages) were computed. A multivariate general linear model (GLM) was used to determine the association between participants' general characteristics and the overall food safety KAP score. Moreover, the correlation between the effect of the COVID-19 pandemic and participants' KAP scores for food safety was measured using the nonparametric Kruskal-Wallis test. Additionally, the test also provided the correlation between major food safety aspect ratings and improvements in participants' food safety measures during COVID-19.

Moreover, a multivariable binary logistic regression analysis was performed using the entry approach to discover variables independently related to a high level of food safety KAP. The characteristics of respondents were factored into the regression models to assess their

relationship with the overall KAP level for food safety. Results with a P-value of < 0.05 were deemed statistically significant. The aggregate of correct responses to the 42 tested questions was used to compute each respondent's overall KAP score for food safety. Each accurate response received one point, while each incorrect answer received zero points. The total KAP scores of respondents were separated into two groups. Respondents who scored more than or equal to 60% of the overall KAP score were thought to have a high level of knowledge, positive attitudes, and appropriate practices. In comparison, less than 60% would be considered inadequate knowledge, negative attitude, and inappropriate practices. The 60% passing rate (cut-off value) was determined by research done by Ayaz, Priyadarshini, and Jaiswal. [28].

## 3. Results and discussion

### 3.1 Respondents' characteristics

A total of 1126 Jordanian women from various regions took part in this study. The respondents aged 24 years or younger represented 18.5% of the total; the respondents aged 25–44 years comprised 55.2% of the total, and the respondents aged 45 years or older comprised 26.6% of the total (Fig 1). The majority of the respondents were married (68.1%), and 31.9% were single. Approximately 15% of those enrolled had a higher university education degree (MSc or Ph.D.), 13.2% had a diploma, 14% were uneducated or had only a 12-year school education, and 57.1% had a Bachelor's degree. Moreover, a third (33.3%) of the participants had fewer than 5 years of experience preparing food, 29.9% had between 5 and 10 years, and 36.8% had more than 10 years. Approximately 72% of respondents said the COVID-19 pandemic raised their concerns about food safety.

### 3.2 Food safety KAP aspects

**3.2.1 Total food safety KAP.**   The overall food safety KAP score was computed by summing the correct responses to the survey's 42 questions. The minimum KAP score for all examined food safety factors was 10, and the maximum was 38 out of 42. This study offers information concerning food safety KAP of women food handlers in households in Jordan during the COVID-19 pandemic. Because of the large sample size (1126), the results are likely representative of the actual status of food safety KAP. The mean total food safety KAP score was 22.1/42, corresponding to 52.6% of the questions answered correctly (Table 1). According to the findings of this study, women food handlers in Jordan had inadequate knowledge, negative attitudes, and inappropriate practices (an overall food safety KAP score of ≤ 60%). The percentage of total correct answers in the current study (52.6%) was lower than that of women in the Sharjah-United Arab Emirates (57.4%) [29]. The current study findings on food safety KAP are similar to those of other work [14, 30–33]. However, the findings of this study disagree with another study conducted in Jordan, in which most consumers in Jordan exhibited good food safety knowledge [17]. This discrepancy in results between the two studies might have been due to differences in the populations studied in each investigation, with only females participating in the current study and both females and males participating in the previous study. Furthermore, the previous study stated that one of its limitations was that the survey questions used did not cover all aspects of food safety knowledge. In contrast, the current study covered almost all aspects of food safety knowledge, attitudes, practices, and COVID-19 [17].

**3.2.2 "Personal hygiene" aspect.**   Correct answers to the "personal hygiene" aspect are presented in Fig 2. They had the highest mean knowledge and practice ratings in personal hygiene, highlighting that women food handlers at homes in Jordan had good knowledge and proper practices about personal hygiene. This is analogous to what has been previously found among Saudi women [34]. Hand washing was prevalent, as predicted. More than 90% of

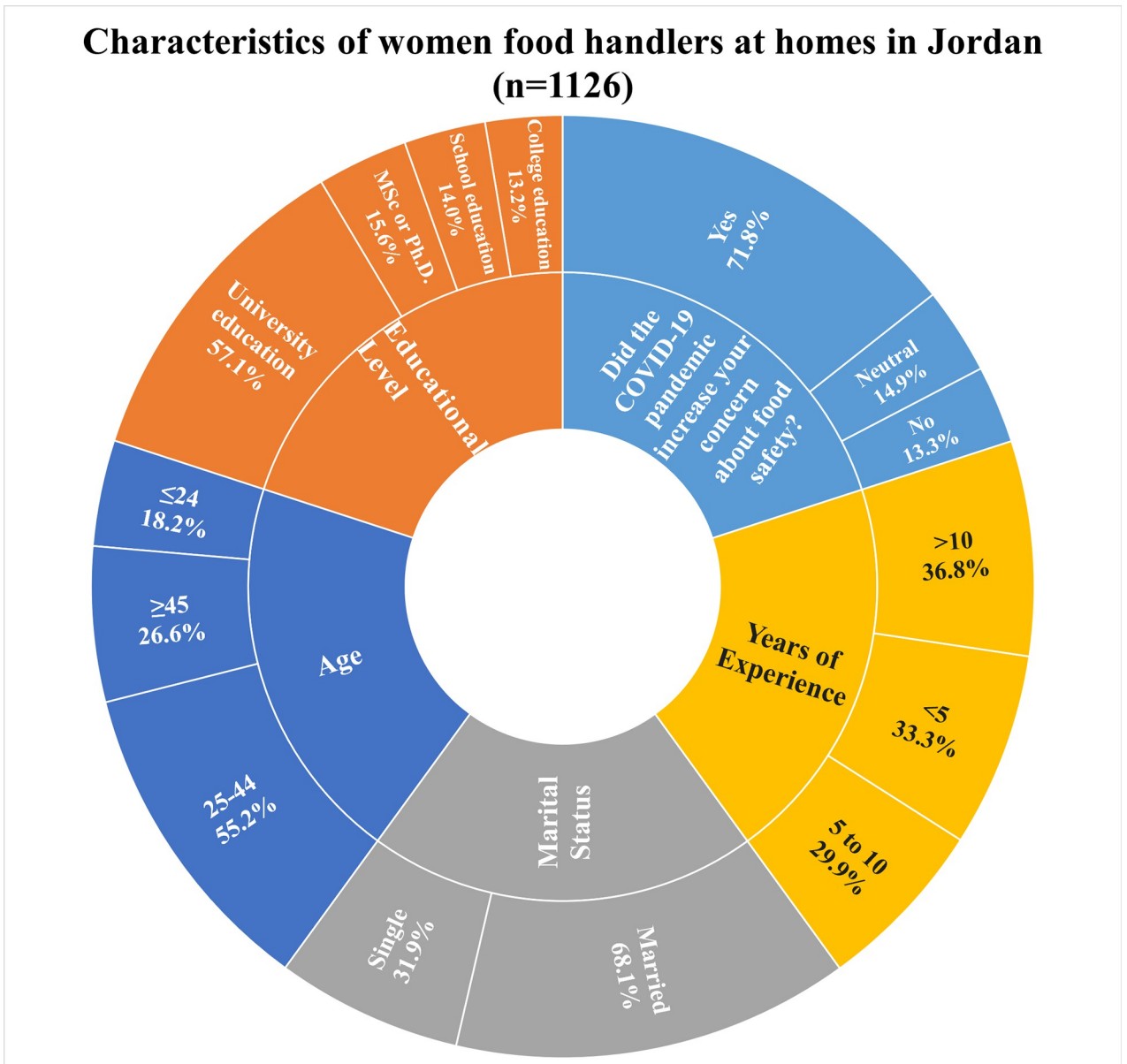

**Fig 1. Characteristics of women food handlers at homes in Jordan (n = 1126).**

respondents could identify 6 situations when handwashing is required, and 38.5% said they wash their hands after using mobile phones. A study conducted among females living in Dubai reported that 57.9% knew that hand washing was not necessary after every stage of food preparation [35]. Moreover, 64.7% of the respondents mentioned knowing how to wash their hands correctly. This is comparable to that reported previously, in which it was stated that 65.3% of Saudi mothers knew the correct manner of handwashing [28]. Also, the current study found that 38.9% of the respondents knew that the duration required for handwashing was 20 sec or more. The Centers for Disease Control and Prevention (CDC) recommend washing hands with hot water and detergent for at least 20 sec to avoid foodborne infections [36].

**Table 1. Food safety KAP score (mean and percentage) of respondents.**

| Aspect | Score Mean ± Std. Error | Score percent[1] Mean ± Std. Error |
|---|---|---|
| Total food safety KAP (42 questions) | 22.1±0.1 | 52.6±0.3 |
| Personal Hygiene (9 questions) | 7.2±0.03 | 79.9±0.4 |
| Cross-contamination prevention (5 questions) | 1.4±0.03 | 28.1±0.6 |
| Cleaning and sanitation (4 questions) | 2.6±0.03 | 64.7±0.6 |
| Safe storage, thawing, cooking, keeping, and reheating of foods (12 questions) | 5.1±0.06 | 42.8±0.5 |
| Health issues that would affect food safety (3 questions) | 1.1±0.02 | 37.9±0.8 |
| Symptoms of foodborne illnesses (5 questions) | 2.6±0.03 | 51.3±0.7 |
| COVID-19 (4 questions) | 2.1±0.03 | 51.8±0.7 |

[1] Score percent = (Score/maximum*100)

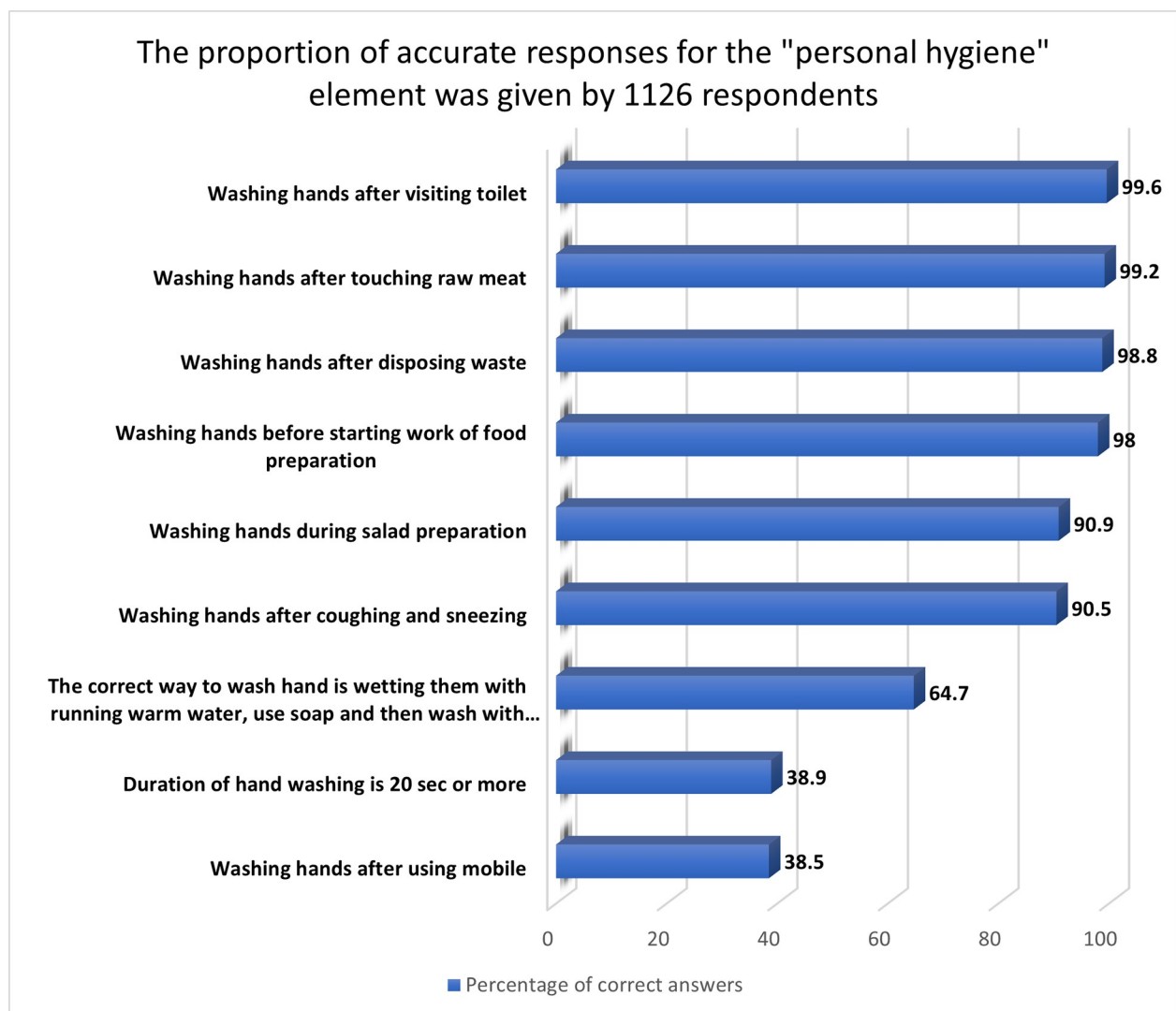

**Fig 2. The proportion of accurate responses for the "personal hygiene" element given by 1126 respondents.**

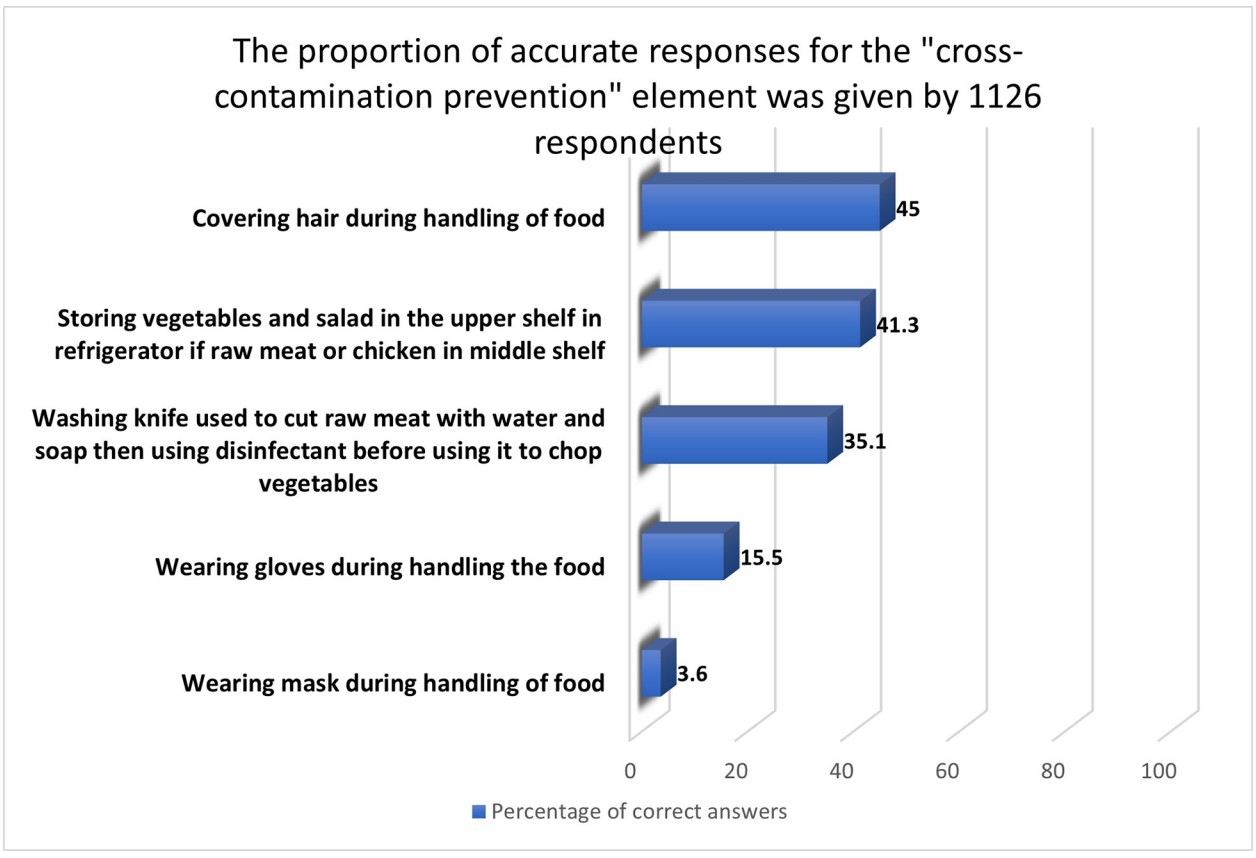

**Fig 3. The proportion of accurate responses for the "cross-contamination prevention" element given by 1126 respondents.**

Other studies have shown that food handlers can have high personal hygiene knowledge and practice [37–39].

**3.2.3 "Cross-contamination prevention" aspect.** The present study found that cross-contamination prevention had the most unfavorable attitudes and incorrect practices, with an average score of 28.1%. Less than half (45%) of those who voted said they cover their hair when handling food (Fig 3). Barely 15.5 and 3.6% of respondents said they use gloves and masks when handling food, respectively. A similar conclusion was reached regarding wearing gloves among food handlers in Egypt's Sohag Governorate (18.6%) [40]. However, 35.1% of respondents correctly responded to a question on preventing food cross-contamination.

Nevertheless, a higher percentage was observed in a study conducted among Canadians [41]. Furthermore, in the current survey, just 41.3% separated ready-to-eat food from raw food in the refrigerator. In contrast, the majority of Sharjah's women (70.1%) do so [29]. A study conducted to assess food safety among consumers at home stated that poor food-handling practices are widespread in homes during food preparation [42].

**3.2.4 "Cleaning and sanitation" aspect.** Cleaning and sanitation knowledge and practices were highly rated, with an average score of 64.7%. The proportion of women who knew how to wipe and sanitize the kitchen countertop (85.4%) was greater than the proportion of Slovenian consumers (59.7%) [32]. Given that the kitchen sink should always be disinfected daily to prevent foodborne diseases [43], about 87% of respondents in the present survey cleaned their kitchen sinks regularly, as shown in Fig 4. This conclusion was remarkably similar to what women in Lahore reported (82.1%) (Naeem, 2018 #305). Conversely, 95.9% of Lahore women

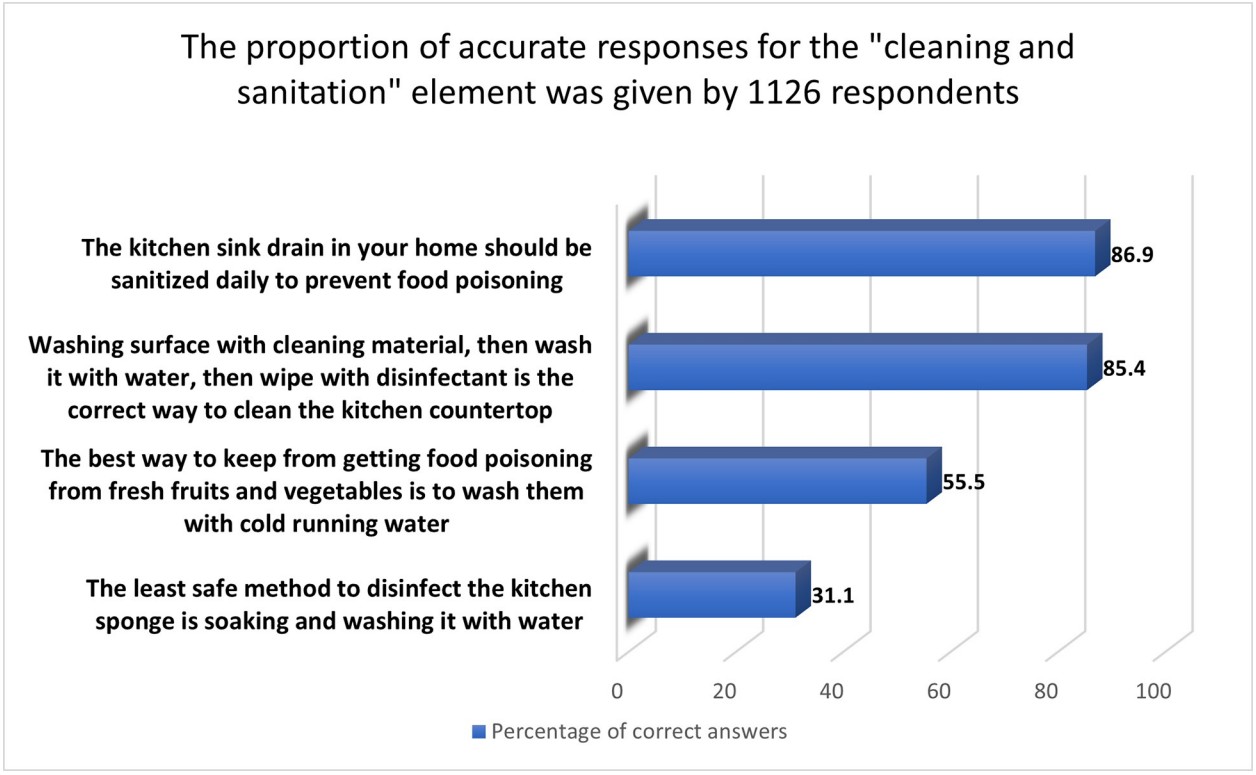

**Fig 4. The proportion of accurate responses for the "cleaning and sanitation" element given by 1126 respondents.**

were aware that washing vegetables and fruits in running cold water made them safe to consume (Naeem, 2018 #305), while in the current study only 55.5% knew that. Finally, just 31.1% said soaking and washing the kitchen sponge in water was the least effective disinfection method. Multiple studies have found kitchen sponges to be a main reservoir of food bacteria that can lead to contamination [44, 45].

**3.2.5 "Safe storage, thawing, cooking, keeping and reheating of foods" aspect.** A low KAP score was obtained among women in the present study about food temperatures, with a mean score of 42.8%, which was similar to that reported in Australia [46] and Ireland [47]. According to the WHO, 45.6% of foodborne illness outbreaks involved temperature abuse during food preparation [48]. Most of the correct answers (87.6%) were about purchasing frozen food at the end of a grocery run and never refreezing defrosted frozen meat, poultry, or fish for later use (Fig 5). These results exceed the one obtained in Solvania [32], in which only 9.9 and 70% of Slovenian consumers knew that frozen food must be purchased at the end of shopping and defrosted raw meat should not be refrozen for later use. A substantial majority of the right answers (70.9%) were about storing leftovers in the fridge for no more than two days. However, just 31.1% women were found to heat the leftovers to boiling temperature properly, and 54.7% quickly discarded unfinished leftovers. The United States Department of Agriculture (USDA) stated that leftovers should be handled according to hygienic procedures, placed in clean dishes, cooled immediately, sealed, and refrigerated at 4–7˚C for no more than three days, or frozen if stored for a longer period. However, women in Lahore had a higher rate of knowledge (90.7%) [49]. Poor knowledge was observed regarding checking whether poultry was sufficiently cooked by its temperature (4.7%), which is similar to that reported among women in Lahore (0%) [11]. Also, negative attitudes were noticed regarding food

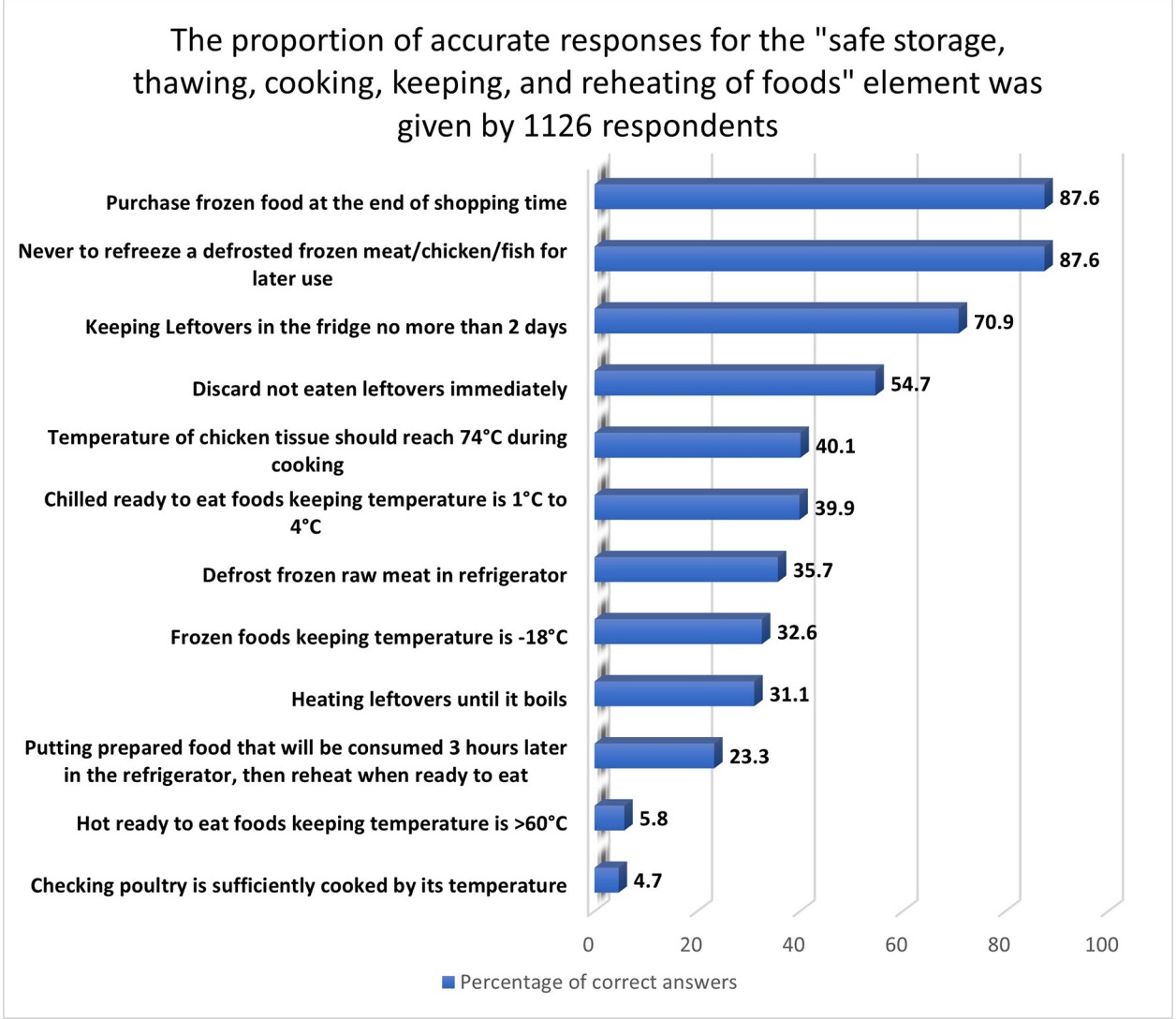

**Fig 5. The proportion of accurate responses for the "safe storage, thawing, cooking, keeping, and reheating of foods" element given by 1126 respondents.**

storage temperatures, in which 39.9, 5.8, 32.6, and 40.1% thought refrigerated ready-to-eat food should indeed be stored at 1–4˚C, heated ready-to-eat food at > 60˚C, frozen food at -18˚C, and chicken meat at 74˚C during cooking. Similarly, research done in the north region of Jordan found that 34, 21, and 33% of female university students, respectively, knew the correct food storage temperatures in the refrigerator, the freezer, and the food's internal temperature during cooking [50]. Finally, only 23.3% of women stated that precooked meals served three hours later should be refrigerated and then warmed when ready to eat.

**3.2.6 "Health issues that would affect food safety" aspect.** With a score percent mean of 37.9%, respondents in our survey had inadequate knowledge and unfavorable attitudes about "health issues that would affect food safety." Pathogenic microorganism-contaminated meals do not have to taste, smell, or appear different from safe-to-eat foods [43], but only 18.2% knew that (Fig 6). In the present study, over two-thirds of the women (63.5%) knew that handling foods with an open cut on the back of their hands was hazardous. This finding indicates

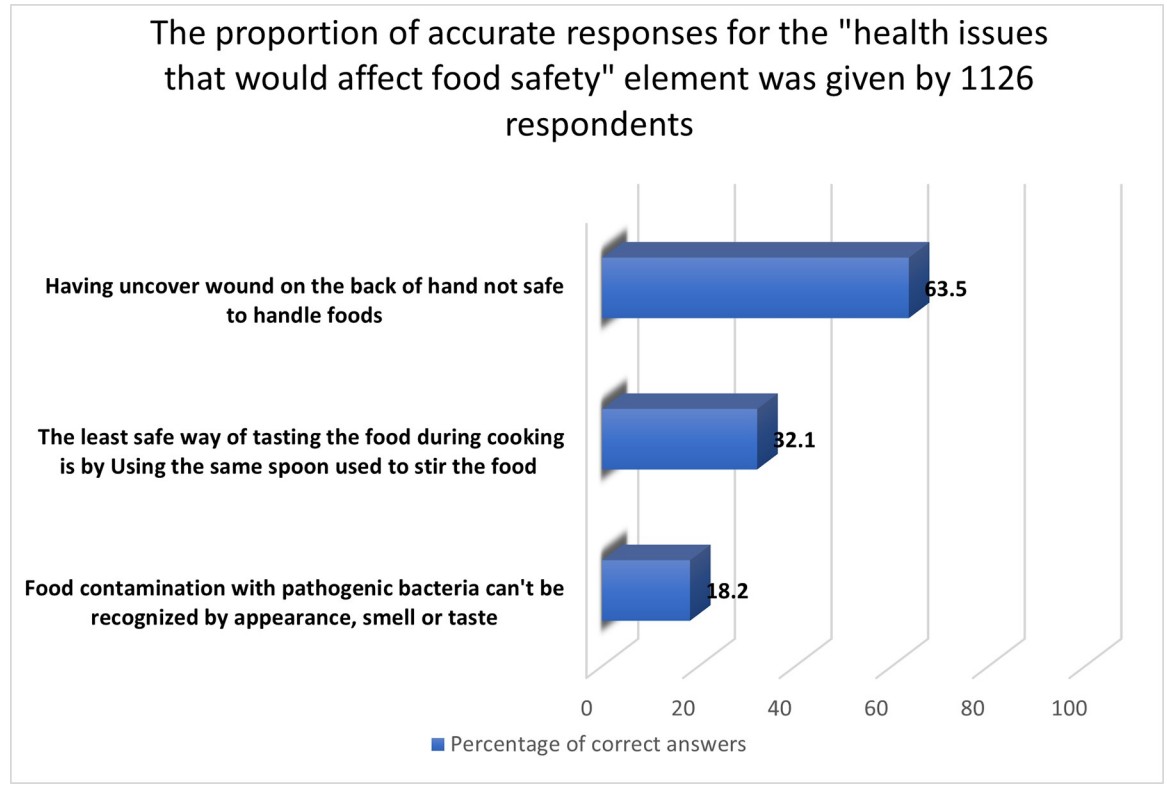

**Fig 6. The proportion of accurate responses for the "health issues that would affect food safety" element was given by 1126 respondents.**

that food handlers in Jordan were more aware of this issue than those in similar studies done in China, where just 25.5% of respondents were aware of the proper response [31], 29.1% in Brazil [51], and 19.6% in Greece [52]. Lastly, 32.1% said they use the cooking utensil to taste food while cooking.

**3.2.7 "Symptoms of foodborne illnesses" aspect.** With a mean score of 51.3 percent, the participants of this study demonstrated poor awareness of foodborne microorganisms and illnesses. Only 26, 24.6, and 37.5 percent of those who voted were aware that hypertension, low blood sugar, and a cold or cough, respectively, are not indicators of foodborne disease (Fig 7). However, the vast majority (98.8%) identified diarrhea and vomiting as symptoms of foodborne disease, which is not surprising given that the most widely mentioned symptoms of foodborne gastroenteritis in the news are diarrhea and vomiting. Finally, 69.4 percent of those who voted said that headaches indicate food poisoning. In Portugal, however, 85 percent of those surveyed could recognize the signs and symptoms of food poisoning [53].

**3.2.8 "COVID-19 KAP" aspect.** Even though 68.4% of participants disinfected food containers well before use or storage, they had inadequate understanding and negative attitudes toward COVID-19 (Fig 8). Only 46.4 and 25.4% of participants knew that the COVID-19 virus could not replicate or be transmitted through food, respectively. About two-thirds (67.1%) of participants thought the COVID-19 virus could persist on rough surfaces for a day (or days). A previous study reported that COVID-19 could persist on inanimate surfaces for 48 to 72 h [54]. Also, WHO concluded that COVID-19 could not proliferate or be transferred by food, but owing to the virus' persistence on surfaces, food may be a vector for virus transmission [55].

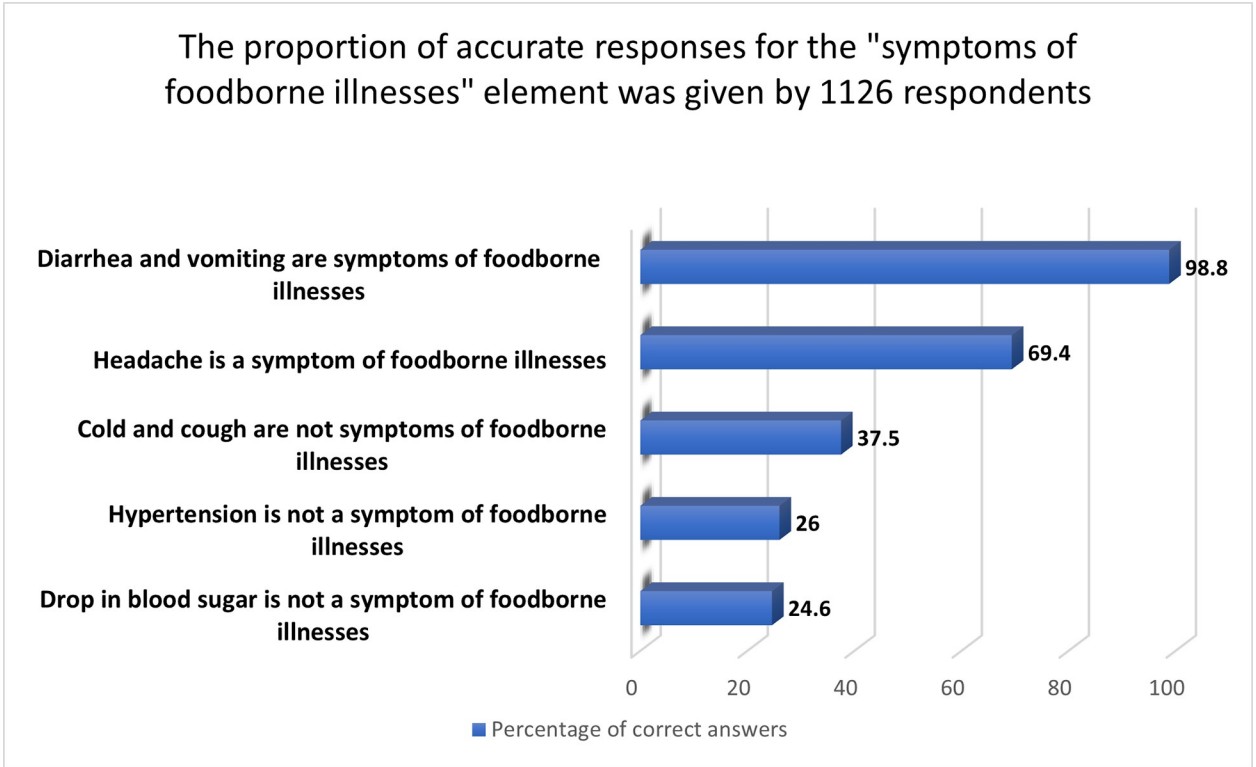

**Fig 7. The proportion of accurate responses for the "symptoms of foodborne illnesses" element given by 1126 respondents.**

### 3.3 Improvements of participants' food safety precautions during COVID-19

The respondents' perspectives on the COVID-19 pandemic and food safety are summarized in Table 2. Most participants (76.7%) were concerned about spreading the COVID-19 virus through the food they provided to others, and 83% were concerned about spreading the infection while preparing meals. In a similar study, 90.1% of the participants among Brazilians indicated that they were more cautious regarding food hygiene practices and attitudes since they thought food might transmit the COVID-19 viral infection [56]. As previously mentioned, the COVID-19 pandemic elevated 71.8% of participants' awareness concerning food safety, but 13.3% said it did not. High percentages (86.3, 92.3, 69.4, and 91.8%) of the respondents revealed that the COVID-19 pandemic improved their food safety precautions by reducing food contamination, maintaining high levels of personal hygiene, managing food temperature, and sanitizing and cleaning food contact areas, respectively.

A similar study stated that the COVID-19 pandemic had affected food handlers' food safety practices in the United States. Hand hygiene, washing food packaging, and using a food thermometer are all examples of this [57]. Lastly, the WHO mobilized people worldwide to increase adherence to hand hygiene to save lives during the COVID-19 pandemic [58].

### 3.4. The relationship between food safety KAP scores and overall respondent characteristics

Social determinants play an essential role in determining consumers' food safety KAP. No significant relationship was observed between marital status and overall food safety KAP (Table 3). Furthermore, significant associations were observed between respondents' overall

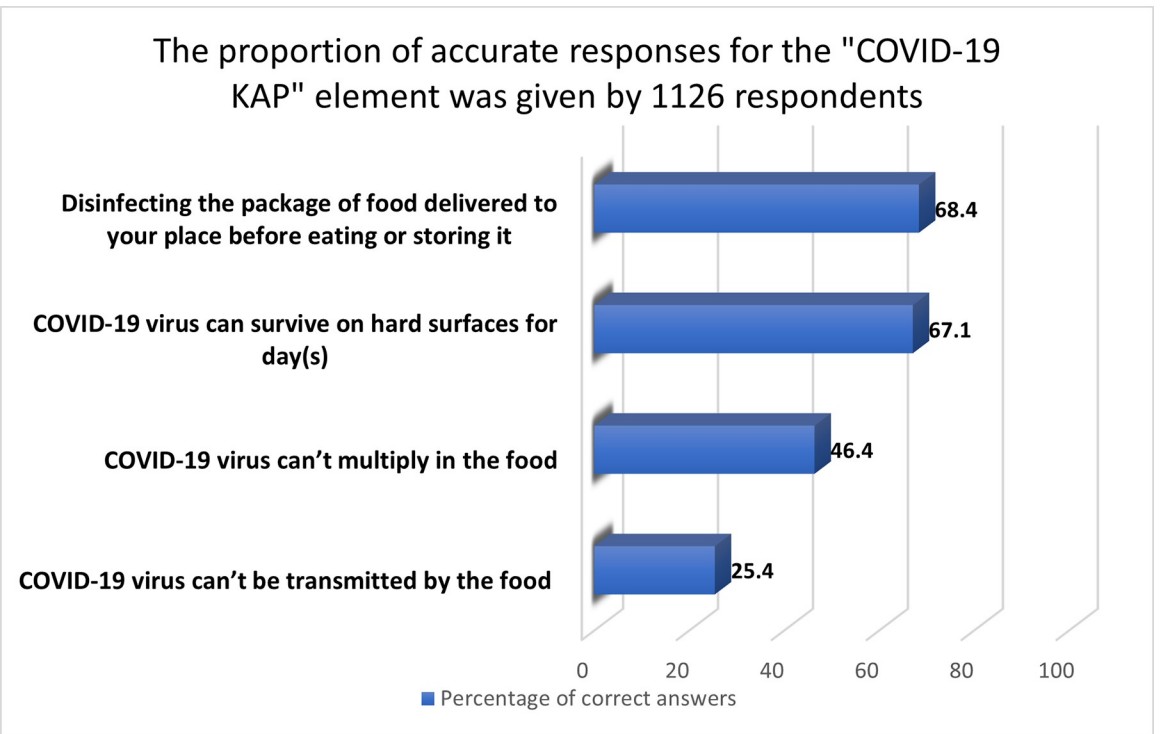

**Fig 8. The proportion of accurate responses for the "COVID-19 KAP" element given by 1126 respondents.**

**Table 2. Query statements and responses on the improvements of participants' food safety precautions during COVID-19.**

| Statement | KAP | Category | Frequency | Percent % |
|---|---|---|---|---|
| • Worrying about transmitting COVID-19 virus infection through the food shared with others | Attitude | Yes | 864 | 76.7 |
| | | Neutral | 143 | 12.7 |
| | | No | 119 | 10.6 |
| • Worrying about transmitting the COVID-19 viral infection to others while preparing food while infected with it | Attitude | Yes | 935 | 83.0 |
| | | Neutral | 87 | 7.7 |
| | | No | 104 | 9.2 |
| • Increasing food safety concerns during the COVID-19 pandemic | Attitude | Yes | 808 | 71.8 |
| | | Neutral | 168 | 14.9 |
| | | No | 150 | 13.3 |
| • Enhancing food safety measures in decreasing food contamination during the COVID-19 pandemic | Attitude | Yes | 972 | 86.3 |
| | | Neutral | 100 | 8.9 |
| | | No | 54 | 4.8 |
| • Enhancing food safety measures in maintaining personal hygiene at a high level during the COVID-19 pandemic | Attitude | Yes | 1039 | 92.3 |
| | | Neutral | 49 | 4.4 |
| | | No | 38 | 3.4 |
| • Enhancing food safety measures in controlling the temperature of food during storage, cooking, keeping, and reheating during the COVID-19 pandemic | Attitude | Yes | 782 | 69.4 |
| | | Neutral | 229 | 20.3 |
| | | No | 115 | 10.2 |
| • Enhancing food safety measures in cleaning and sanitation of food contact surfaces during the COVID-19 pandemic | Attitude | Yes | 1034 | 91.8 |
| | | Neutral | 57 | 5.1 |
| | | No | 35 | 3.1 |

**Table 3. The relationship between food safety KAP scores and overall respondent characteristics.**

| Respondents' characteristics | | Score Mean ± Std. Error | Score percent[1] Mean ± Std. Error | P-value |
|---|---|---|---|---|
| Age[2] | | | | |
| | ≤24 | 21.6[b]±0.3 | 51.5±0.8 | **0.048** |
| | 25–44 | 22.0[ab]±0.2 | 52.5±0.4 | |
| | ≥45 | 22.5[a]±0.2 | 53.7±0.6 | |
| Educational Level[2] | | | | |
| | Uneducated or a 12-year school education | 21.25[b]±0.3 | 50.6±0.8 | **0.001** |
| | Diploma degree | 21.95[ab]±0.3 | 52.3±0.7 | |
| | Bachelor degree | 22.07[b]±0.2 | 52.6±0.4 | |
| | Higher university education (MSc or Ph.D.) | 23.07[a]±0.4 | 54.9±0.8 | |
| Marital Status[2] | | | | |
| | Single | 21.8±0.2 | 51.8±0.6 | 0.06 |
| | Married | 22.3±0.1 | 53.0±0.3 | |
| Experience[2] | | | | |
| | <5 | 21.3[b]±0.2 | 50.7±0.5 | **0.00** |
| | 5–10 | 22.3[a]±0.2 | 53.0±0.5 | |
| | >10 | 22.7[a]±0.2 | 54.1±0.4 | |
| Region[2] | | | | |
| | North | 21.8[b]±0.2 | 51.9±0.5 | **0.03** |
| | Central | 22.3[a]±0.2 | 53.2±0.4 | |
| | South | 21.4[b]±0.4 | 50.8±1.0 | |
| Increased food safety concerns during COVID-19 [3] | | | | |
| | Yes | 22.2[a]±0.1 | 52.7±0.3 | **0.03** |
| | Neutral | 21.5[b]±0.4 | 51.2±0.9 | |
| | No | 22.4[a]±0.4 | 53.4±0.9 | |

[1] Score percent equal to (Score/maximum*100)

[2] Using multivariate general linear model (GLM).

[3] Using Kruskal-Wallis Test.

Note: At P ≤ 0.05, different letters for means within the same categories differ significantly. All factors were adjusted for one another.

food safety KAP scores and age, educational level, experience, and the effect of the COVID-19 pandemic on food safety concerns (P < 0.05). The results showed that respondents had significantly higher food safety KAP scores if they were more than 44 years old (22.5%) compared to those less than 24 years (21.6%). Also, respondents who had a higher university education (MSc or Ph.D.) (23.07%) had significantly better food safety KAP scores than those who had a bachelor's degree (22.07%), were uneducated or had a 12-year school education (21.25%). Respondents had significantly better food safety KAP scores with more than 10 years' experience (22.7%) and within 5–10 years' experience (22.3%) compared to respondents with less than 5 years' experience (21.3%). Lastly, respondents who said the COVID-19 pandemic increased their concern about food safety (22.2%) and did not (22.4%) had received significantly higher KAP ratings for food safety than individuals who were neutral about that (21.5%). Similarly, there was a significant relationship between total food safety attitudes with each gender, age, educational level, and the degree of attention paid to the COVID-19 pandemic situation in a study conducted in China [21]. Previous studies have demonstrated significant differences between food safety KAP and demographics, such as gender, age, race, and income [59]. Also, a study reported significant differences in gender, marital status, educational level, and occupation [14].

## 3.5 The impact of demographic variables on food safety KAP scores using logistic regression analysis

Scores were separated into groups; (1) scores more than or equal to 60%, (2) scores less than 60%, as the 60% cut-off point was (25/42). All variables were adjusted for each other. The food safety KAP score of the participants significantly ($P < 0.05$) affected by experience; respondents with less than 5 years' experience and those with 5–10 years' experience were 1.91 and 1.53 times more likely to have a significantly lower food safety KAP score (OR 1.91 and 1.53) than those with 10 years' experience (Table 4). These findings are supported by many studies [34, 53, 60–62]. As expected, the food safety KAP score of the participants were also significantly (*P < 0.05*) affected by educational level; respondents who had a diploma and were uneducated or had only school education ($\leq$ 12 years) were 2.39 and 2.16 times more likely to have significantly lower food safety KAP score (OR 2.39 and 2.16) than those who a had higher university education (MSc or Ph.D.). Similarly, a study conducted among students in East Malaysia used a binary logistic test to determine the impact of socio-demographic variables on food safety knowledge. It found that educational level was one of the predictors of food safety knowledge [16].

**Table 4. Food safety KAP predictors using logistic regression analysis.**

| Variable [1] | | OR (CI$_{95}$) | P-value |
|---|---|---|---|
| **Age** | | | |
| | $\leq$24 | 0.995 (0.55,1.80) | 0.998 |
| | 25–44 | 1.16 (0.81,1.66) | 0.42 |
| | $\geq$45 | Reference | |
| Educational Level | | | |
| | Uneducated or a 12-year school education | 2.16 (1.22,3.84) | 0.01* |
| | Diploma degree | 2.39 (1.33,4.29) | 0.00* |
| | Bachelor degree | 1.32 (0.88,1.96) | 0.18 |
| | Higher university education (MSc or Ph.D.) | Reference | |
| Marital Status | | | |
| | Single | 0.84 (0.57,1.25) | 0.39 |
| | Married | Reference | |
| Experience | | | |
| | <5 | 1.91 (1.24,2.94) | 0.00* |
| | 5–10 | 1.53 (1.05,2.22) | 0.03* |
| | >10 | Reference | |
| Region | | | |
| | North | 0.89 (0.44,1.80) | 0.74 |
| | Central | 0.67 (0.34,1.33) | 0.25 |
| | South | Reference | |
| Increased food safety concerns during COVID-19 | | | |
| | Yes | 1.05 (0.68,1.62) | 0.82 |
| | Neutral | Reference | |
| | No | 0.78 (0.45,1.35) | 0.38 |

[1] All variables were adjusted to each other.

* P-value $\leq$ 0.05 was considered statistically significant

### 3.6 The correlation between major food safety aspect ratings and improvements in participants' food safety measures during COVID-19

No significant correlation was observed between the "cross-contamination prevention" and "personal hygiene" aspect scores and the participants' attitudes toward decreasing food contamination and maintaining good personal hygiene during the COVID-19 pandemic (Table 5). On the other hand, a significant correlation between the "cleaning and sanitation" aspect score and the participants' attitudes toward cleaning and sanitation of food contact surfaces during the COVID-19 pandemic ($P < 0.05$) was observed. Participants who thought their precautions for cleaning and sanitation of food contact surfaces were enhanced (2.6%) and those who did not think so (2.9%) had a significantly higher score than those who were neutral (2.2%). Also, significant correlations were observed between the aspect score of "safe storage, thawing, cooking, keeping, and reheating of food" and the participants' attitudes toward controlling the temperature of food during the COVID-19 pandemic ($P < 0.05$). The results indicated that the participants who thought their precautions were enhanced during the COVID-19 pandemic (5.2%) had a significantly greater score than those who did not (4.7%). In Brazil, 72% desired to learn more about sanitizing food and cleaning food packaging and were more concerned about the safety and hygiene of food contact surfaces [56]. Moreover, the COVID-19 pandemic has led to considerable changes in food purchasing, handling,

**Table 5. The correlation between major food safety aspect ratings and improvements in participants' food safety measures during COVID-19.**

| Aspect | COVID-19 statement | | Score Mean ± Std. Error | Score percent[1] Mean ± Std. Error | P-value |
|---|---|---|---|---|---|
| Food contamination prevention | Enhancing food safety measures in decreasing food contamination during the COVID-19 pandemic | | | | |
| | | Yes | 2.6±0.0 | 64.8±0.7 | 0.73 |
| | | Neutral | 2.5±0.1 | 62.3±2.2 | |
| | | No | 2.7±0.1 | 67.6±3.1 | |
| Personal Hygiene | Enhancing food safety measures in maintaining high personal hygiene during the COVID-19 pandemic | | | | |
| | | Yes | 7.2±0.03 | 79.9±0.4 | 0.53 |
| | | Neutral | 7.0±0.2 | 78.0±2.0 | |
| | | No | 7.3±0.2 | 81.6±1.8 | |
| Cleaning and sanitation | Enhancing food safety measures in cleaning and sanitation of food contact surfaces during the COVID-19 pandemic | | | | |
| | | Yes | 2.6[a]±0.03 | 65.0±0.65 | **0.02** |
| | | Neutral | 2.2[b]±0.14 | 55.7±3.43 | |
| | | No | 2.9[a]±0.15 | 72.1±3.66 | |
| Safe storage, thawing, cooking, keeping, and reheating of foods | Enhancing food safety measures in controlling the temperature of food during storage, cooking, keeping, and reheating during the COVID-19 pandemic | | | | |
| | | Yes | 5.2[a]±0.07 | 43.6±0.6 | **0.01** |
| | | Neutral | 5.1[ab]±0.13 | 42.1±1.1 | |
| | | No | 4.7[b]±0.20 | 39.0±1.7 | |

[1] Score percent equal to (Score/maximum*100)

Note: At $P \leq 0.05$, different letters for means within the same categories differ significantly. All factors were adjusted for one another.

and hygiene practices in Lebanon, Jordan, and Tunisia [63]. Therefore, it is vital to conduct continuous comprehensive education and training programs for domestic food handlers, especially women, to protect themselves and their families against foodborne illnesses.

## 4. Conclusion

The present study reveals that during the COVID-19 pandemic, women handling food at home in Jordan had limited knowledge, unfavorable attitudes, and unsafe food handling practices. Therefore, engaging the public in continuous education regarding proper food handling practices is crucial. Food safety efforts are often focused on food supply chains, with little attention paid to food handling and practices in the home. Hence, domestic food handlers require comprehensive education and training to protect family members from foodborne illnesses.

## 5. Limitation

This study could face some limitations as it examined self-reported food safety practices, which are prone to bias by respondents. Also, questions in the demographics part might not cover all social determinants that could impact food safety KAP, such as income, number of family members, and occupation.

## Acknowledgments

The authors would like to thank volunteers for their assistance in providing relevant information during the research.

## Author Contributions

**Conceptualization:** Anas A. Al-Nabulsi, Tareq M. Osaili, Amin N. Olaimat.

**Data curation:** Tasneem M. Al-Jaberi, Sawsan Mutlaq.

**Formal analysis:** Tasneem M. Al-Jaberi.

**Funding acquisition:** Anas A. Al-Nabulsi, Tareq M. Osaili, Amin N. Olaimat.

**Investigation:** Tasneem M. Al-Jaberi, Sawsan Mutlaq.

**Methodology:** Anas A. Al-Nabulsi, Tareq M. Osaili.

**Project administration:** Anas A. Al-Nabulsi, Tareq M. Osaili.

**Resources:** Tasneem M. Al-Jaberi, Sawsan Mutlaq.

**Supervision:** Anas A. Al-Nabulsi, Tareq M. Osaili.

**Validation:** Anas A. Al-Nabulsi, Tareq M. Osaili, Amin N. Olaimat.

**Visualization:** Tasneem M. Al-Jaberi.

**Writing – original draft:** Tasneem M. Al-Jaberi.

**Writing – review & editing:** Anas A. Al-Nabulsi, Tareq M. Osaili, Amin N. Olaimat.

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
