## [Decision Letter · Decision Letter 0]

12 Apr 2023

PONE-D-22-35672Food safety literacy among women food handlers at homes in Jordan during the COVID-19 pandemicPLOS ONE

Dear Dr. Osaili,

Thank you for submitting your manuscript to PLOS ONE. After careful consideration, we feel that it has merit but does not fully meet PLOS ONE’s publication criteria as it currently stands. Therefore, we invite you to submit a revised version of the manuscript that addresses the points raised during the review process.

Your manuscript has been assessed by three expert reviewers, whose comments are appended below and in the attached documents. The reviewers have highlighted concerns about several aspects of the methodological reporting and the theoretical basis of some decisions. Please ensure you respond to each point carefully in your response to reviewers document, and modify your manuscript accordingly. The manuscript also needs to be proofread carefully to ensure that the revised version is written in clear, grammatical and unambiguous English.

We look forward to receiving your revised manuscript.

Kind regards,

Dr Joseph Donlan

Senior Editor

PLOS ONE

Journal Requirements:

"The authors would like to thank Jordan University of Science and Technology's deanship of research for financing this effort..'"

**Additional Editor Comments:**

Reviewer 1:

It is a well planned study to assess the food safety literacy among women food handlers at homes in Jordan.

I have few questions. How the sample size was calculated? The designed questionnaire was in which language? As it is mentioned that questionnaire was distributed on social media, how it was ensured that it was rightly filled by a house woman? Please mention the necessary actions that can be taken on the basis of findings of this study to improve the knowledge of food safety.

Reviewer 2:

See the attached word,, Specific comments, as well as the revised pdf manuscript showing notes.

See also below:Introduction

Page 3, Lines 56-58. What about studies conducted in developing countries, related to women food handlers? You should provide examples, if available, related to women food handling, in the Middle East area, and what data showed.

Page 3, Lines 56-59. The EU studies, have these shown that the FBIOs are related to women food handlers? At home, be more specific. What it food prepared at home, or purchased from outlets other than home, or badly handled at home (that is poor personal hygiene). I do not see how this study relates to women food handlers… unless you provide more details. .

Page 3, Line 63. Did any of these studies, involve women? If any of those, did, what were the results? In summary? Expand on 1 or 2 studies…

Page 3, Line 64. Do you perhaps mean “regional” instead of local? And if local is meant in Jordan, in what is your study being different from previous? And what were the results from these studies? If you meant, regional like Lebanon etc… state and give a short paragraph on these results.

Page 3, Lines 68-80. Move this paragraph, in the beginning of the Introduction, best after line 41.

Page 4, Line 73-76. Delete lines… that is…starting from “Most to situation”.

Page 4, Line 81. Reword “Confront” to confronted… Use also past tense in other cases…in text, such as line 83; There “was” an urgent need…

Pages 4-5, Line 83-87. Too long paragraph and needs splitting in 2 shorter sentences.

Page 5, Line 86. “KAP” of what?

Pages 4-5, Line 83-89. Are there any other related data on women food handlers, regional or global, and what the data showed? Include a paragraph, if available, to justify the need of limited regional or global studies, on women, especially handling food in the Middle East.

Materials and Methods

Page 5, Line 103. Capitalize, “the Dep. of Statistics”

Page 6, Line 122. Rewrite in these two parts…. In the second and third part.

Page 6, Line 126. Reword “accurately assess what was meant to be assessed”. Why was a validation was conducted to those interviewed “face to face”?

Page 6, Line 127. Replace people with “women” and the word “subjects” throughout the manuscript, also Line Page 7, Lines 149, 150 etc.

Page 7, Line 140. Correct “Not” to not.

Page 8, Lines 164-165. Have other studies used a different % (60) value? And why did you decide to use the 60% value, based on 1 study only? Justify.

Results and Discussion

Page 8, Lines 168. In all, 1126 Jordanians. Reword better to “Jordanian” women…

Page 8, Lines 172, 174. Replace the word “polled” with voted, as this is a word used for elections, and in other sentences.

Page 9, Lines 186-187. “women food handlers in Jordan had inadequate knowledge, negative attitudes, and inappropriate practices”…. Is there a reason/reasons for that? Can you provide, why this is the case?

Page 10, Line 205 . Replace the word “discovered” with reported or corroborated.

Reword Legends of Figs 2-8. “Was given”.. Does not sound correct, perhaps reword with “given by”…

Page 11, Line 229. Did the “Canadian” study involve women food handlers? If yes, include it to validate your results.

Page 11, Line 230-231. “just 41.3 % segregate ready-to-eat food from raw food in the

231 refrigerator”, you meant here “separate” or reword.

Page 12, Line 237. Reword “Slovenian customers”.

Page 12, Line 238. Edit the word “About” to “about”.

Page 12, Lines 240, 245. Edit the word “identical” to “similar”, and food germs to “food bacteria”.

Page 13, Line 255. “These findings are better than “that”. Reword these 2 words.

Page 13, Lines 259, 263, 268. “of household women heat leftovers until they are boiling”. Reword. Also, “however, women in Lahore had a higher rate”. Include “of knowledge”. Also, and chicken tissue with “chicken meat”, and in Jordan's north… does not make sense. Page 14, Line 284. “tasting meals with the same utensils used to mix them”. Reword.

Pages 14, 15…... Lines 388, 387, 383, 365, 360, 309, 301, 302, 299, 288, 289, 294. “subjects and polled”. Reword.

Page 18, Lines 358, 369. “60th percentile”, should that be 60% or percentage?

Page 20, Lines 378, 381, 385. Perhaps “correlations”? instead of connections?

Discussion: Well supported, reasoned and being comprehensive.

Tables: Well presented, as well also Figures, with statistical evaluations included.

Conclusions: Adequate and well drawn.

*“Please also refer to the revised Pdf copy” for additional corrections.

Reviewer 3:

Some minor corrections:

page 9, line 191, some references were mentioned directly. Corrections should be done for all the references.

Page 11, line 217, Figures mentioned but there were no figures in all the manuscript.

Reviewers' comments:

Reviewer's Responses to Questions

**Comments to the Author**

1. Is the manuscript technically sound, and do the data support the conclusions?

Reviewer #1: Yes

Reviewer #2: Yes

Reviewer #3: Yes

2. Has the statistical analysis been performed appropriately and rigorously? 

Reviewer #1: Yes

Reviewer #2: Yes

Reviewer #3: Yes

3. Have the authors made all data underlying the findings in their manuscript fully available?

Reviewer #1: Yes

Reviewer #2: Yes

Reviewer #3: Yes

4. Is the manuscript presented in an intelligible fashion and written in standard English?

Reviewer #1: Yes

Reviewer #2: Yes

Reviewer #3: Yes

5. Review Comments to the Author

Reviewer #1: It is a well planned study to assess the food safety literacy among women food handlers at homes in Jordan.

I have few questions. How the sample size was calculated? The designed questionnaire was in which language? As it is mentioned that questionnaire was distributed on social media, how it was ensured that it was rightly filled by a house woman? Please mention the necessary actions that can be taken on the basis of findings of this study to improve the knowledge of food safety.

Reviewer #2: See the attached word,, Specific comments, as well as the revised pdf manuscript showing notes.

See also below:Introduction

Page 3, Lines 56-58. What about studies conducted in developing countries, related to women food handlers? You should provide examples, if available, related to women food handling, in the Middle East area, and what data showed.

Page 3, Lines 56-59. The EU studies, have these shown that the FBIOs are related to women food handlers? At home, be more specific. What it food prepared at home, or purchased from outlets other than home, or badly handled at home (that is poor personal hygiene). I do not see how this study relates to women food handlers… unless you provide more details. .

Page 3, Line 63. Did any of these studies, involve women? If any of those, did, what were the results? In summary? Expand on 1 or 2 studies…

Page 3, Line 64. Do you perhaps mean “regional” instead of local? And if local is meant in Jordan, in what is your study being different from previous? And what were the results from these studies? If you meant, regional like Lebanon etc… state and give a short paragraph on these results.

Page 3, Lines 68-80. Move this paragraph, in the beginning of the Introduction, best after line 41.

Page 4, Line 73-76. Delete lines… that is…starting from “Most to situation”.

Page 4, Line 81. Reword “Confront” to confronted… Use also past tense in other cases…in text, such as line 83; There “was” an urgent need…

Pages 4-5, Line 83-87. Too long paragraph and needs splitting in 2 shorter sentences.

Page 5, Line 86. “KAP” of what?

Pages 4-5, Line 83-89. Are there any other related data on women food handlers, regional or global, and what the data showed? Include a paragraph, if available, to justify the need of limited regional or global studies, on women, especially handling food in the Middle East.

Materials and Methods

Page 5, Line 103. Capitalize, “the Dep. of Statistics”

Page 6, Line 122. Rewrite in these two parts…. In the second and third part.

Page 6, Line 126. Reword “accurately assess what was meant to be assessed”. Why was a validation was conducted to those interviewed “face to face”?

Page 6, Line 127. Replace people with “women” and the word “subjects” throughout the manuscript, also Line Page 7, Lines 149, 150 etc.

Page 7, Line 140. Correct “Not” to not.

Page 8, Lines 164-165. Have other studies used a different % (60) value? And why did you decide to use the 60% value, based on 1 study only? Justify.

Results and Discussion

Page 8, Lines 168. In all, 1126 Jordanians. Reword better to “Jordanian” women…

Page 8, Lines 172, 174. Replace the word “polled” with voted, as this is a word used for elections, and in other sentences.

Page 9, Lines 186-187. “women food handlers in Jordan had inadequate knowledge, negative attitudes, and inappropriate practices”…. Is there a reason/reasons for that? Can you provide, why this is the case?

Page 10, Line 205 . Replace the word “discovered” with reported or corroborated.

Reword Legends of Figs 2-8. “Was given”.. Does not sound correct, perhaps reword with “given by”…

Page 11, Line 229. Did the “Canadian” study involve women food handlers? If yes, include it to validate your results.

Page 11, Line 230-231. “just 41.3 % segregate ready-to-eat food from raw food in the

231 refrigerator”, you meant here “separate” or reword.

Page 12, Line 237. Reword “Slovenian customers”.

Page 12, Line 238. Edit the word “About” to “about”.

Page 12, Lines 240, 245. Edit the word “identical” to “similar”, and food germs to “food bacteria”.

Page 13, Line 255. “These findings are better than “that”. Reword these 2 words.

Page 13, Lines 259, 263, 268. “of household women heat leftovers until they are boiling”. Reword. Also, “however, women in Lahore had a higher rate”. Include “of knowledge”. Also, and chicken tissue with “chicken meat”, and in Jordan's north… does not make sense. Page 14, Line 284. “tasting meals with the same utensils used to mix them”. Reword.

Pages 14, 15…... Lines 388, 387, 383, 365, 360, 309, 301, 302, 299, 288, 289, 294. “subjects and polled”. Reword.

Page 18, Lines 358, 369. “60th percentile”, should that be 60% or percentage?

Page 20, Lines 378, 381, 385. Perhaps “correlations”? instead of connections?

Discussion: Well supported, reasoned and being comprehensive.

Tables: Well presented, as well also Figures, with statistical evaluations included.

Conclusions: Adequate and well drawn.

*“Please also refer to the revised Pdf copy” for additional corrections.

Reviewer #3: Some minor corrections:

page 9, line 191, some references were mentioned directly. Corrections should be done for all the references.

Page 11, line 217, Figures mentioned but there were no figures in all the manuscript.

6. PLOS authors have the option to publish the peer review history of their article (what does this mean?). If published, this will include your full peer review and any attached files.

Reviewer #1: **Yes: **Hafiz Muhammad Shahbaz

Reviewer #2: No

Reviewer #3: No

---

## [Author Response · Author response to Decision Letter 0]

26 May 2023

The manuscript has been revised to reflect all the comments. The "Response to Reviewers' file is now attached to the files section.

---

## [Decision Letter · Decision Letter 1]

26 Jun 2023

Food Safety Knowledge, Attitudes, and Practices Among Jordanian Women Handling Food at Home During COVID-19 Pandemic

PONE-D-22-35672R1

Dear Dr. Osaili,

We’re pleased to inform you that your manuscript has been judged scientifically suitable for publication and will be formally accepted for publication once it meets all outstanding technical requirements.

Kind regards,

Ayechew Ademas Tesema, MSc

Academic Editor

PLOS ONE

Additional Editor Comments (optional):

Reviewers' comments:

Reviewer's Responses to Questions

**Comments to the Author**

1. If the authors have adequately addressed your comments raised in a previous round of review and you feel that this manuscript is now acceptable for publication, you may indicate that here to bypass the “Comments to the Author” section, enter your conflict of interest statement in the “Confidential to Editor” section, and submit your "Accept" recommendation.

Reviewer #1: All comments have been addressed

Reviewer #3: All comments have been addressed

2. Is the manuscript technically sound, and do the data support the conclusions?

Reviewer #1: Yes

Reviewer #3: Yes

3. Has the statistical analysis been performed appropriately and rigorously? 

Reviewer #1: Yes

Reviewer #3: Yes

4. Have the authors made all data underlying the findings in their manuscript fully available?

Reviewer #1: Yes

Reviewer #3: Yes

5. Is the manuscript presented in an intelligible fashion and written in standard English?

Reviewer #1: Yes

Reviewer #3: Yes

6. Review Comments to the Author

Reviewer #1: The authors have addressed my concerns and made the manuscript acceptable for publication. The abstract is comprehensive. Methodology followed is correct. The data is correctly analysed.

Reviewer #3: Good Job and all the corrections have been done according to our suggestions. Best wishes for the authors

7. PLOS authors have the option to publish the peer review history of their article (what does this mean?). If published, this will include your full peer review and any attached files.

Reviewer #1: **Yes: **Hafiz Muhammad Shahbaz

Reviewer #3: No

---

## [Editor Report · Acceptance letter]

28 Jun 2023

PONE-D-22-35672R1 

Food Safety Knowledge, Attitudes, and Practices Among Jordanian Women Handling Food at Home During COVID-19 Pandemic 

Dear Dr. Osaili:

I'm pleased to inform you that your manuscript has been deemed suitable for publication in PLOS ONE. Congratulations! Your manuscript is now with our production department. 

Kind regards, 

on behalf of

Assistant professor Ayechew Ademas Tesema 

Academic Editor

PLOS ONE